# Functionalized GO/Hydroxy-Terminated Polybutadiene Composites with High Anti-Migration and Ablation Resistance Performance

**DOI:** 10.3390/polym14163315

**Published:** 2022-08-15

**Authors:** Shuai Yin, Zhehong Lu, Haoran Bai, Xinyang Liu, Hao Li, Yubing Hu

**Affiliations:** 1National Special Superfine Powder Engineering Research Center of China, Nanjing University of Science and Technology, Nanjing 210014, China; 2College of Materials Science and Engineering, Shenyang University of Technology, Shenyang 110870, China; 3College of Materials Science and Engineering, Nanjing Tech University, Nanjing 211816, China

**Keywords:** graphene oxide, ablation resistance, propellant liner, anti-migration

## Abstract

The migration of plasticizers such as nitroglycerin seriously affects the storage and working safety of rocket systems. In this work, hydroxy-terminated polybutadiene (HTPB) liner composites with the cross-linked structure were prepared by cross-linking isocyanate functionalized graphene oxide (IGO) with HTPB to prevent the migration of high energy plasticizers in the propellant. IGO was uniformly dispersed in the matrix as reinforcement and cross-linker, providing good migration resistance and ablation resistance for the liner composites. Compared with pure HTPB, the migration resistance of the liner with 0.5 wt% IGO increased by 18.94%, 16.33% and 15.34% at 25 °C, 60 °C and 90 °C, respectively. In addition, the ablation resistance of the HTPB liner was improved by the addition of IGO. The improved anti-migration properties come from the special laminar structure of IGO and the dense molecular chains network of the cross-linked composites.

## 1. Introduction

The propellant liner is a viscoelastic substance that firmly bonds the propellant grain and the inhibitor layer [1,2]. It is an indispensable and important part of the shell-bonded solid rocket motor [3,4]. Environmental factors such as external temperature, light, moisture, and water can cause the material itself to age during the solid motor storage. Small molecule plasticizers and nitroglycerine (NG), which are abundant in propellants, can diffuse and migrate from the propellant column to the inhibitor during storage, seriously damaging the interface between the inhibitor and the propellant [5,6,7]. In addition, the migration of large amounts of NG may damage the solid engine and cause major accidents [8,9,10]. Finding effective methods to address the problem of small molecule migration in propellant in terms of material design and mechanism is a key scientific problem that needs to be solved urgently [11,12,13].

Many works have attempted to prevent the migration of NG; the most convenient method is using a polymer with strong anti-migration ability as the inhibitor [2,14]. EPDM rubber has strong resistance to small molecules and Lu Z.H. et al., used it to resist dioctyl sebacate (DOS) migration. The addition of inorganic filler is also beneficial to the anti-migration performance, as inorganic filler can increase the density of the coating layer to play an anti-migration role [15]. In addition, setting a barrier layer between the propellant and the inhibitor can significantly reduce the migration of NG [16]. Wu Wei et al. [16] used PF, PVF and KH550 to prepare an anti-migration barrier layer for a dual-base propellant and investigated the effect of the barrier layer on the migration of NG by accelerated migration experiments; they concluded that a barrier layer material with high cross-link density and low polarity could better prevent the migration of NG. HTPB is a translucent liquid rubber that emerged in the 1960s, with good transparency and low glass transition temperature [17,18,19,20]. In addition, it can be cured and cross-linked at room temperature or high temperature to form an elastomer with a three-dimensional network structure, which has excellent mechanical properties and good corrosion resistance, making it suitable for solid propellant liner in the aerospace industry [21]. Isophorone diisocyanate (IPDI) and other isocyanates can react with HTPB to form urethane networks, which are widely used in industry. Li et al. [22] made a HTPB/TGO liner, which significantly improved the migration resistance of DOS.

Graphene is a two-dimensional carbon nanomaterial consisting of carbon atoms in a hexagonal honeycomb lattice with sp^2^ hybrid orbitals [23,24]. The electron density of the internal benzene ring is so high that neither atoms nor molecules can penetrate it, making graphene excellent in resisting small molecule penetration [25,26]. As a derivative of graphene, GO has obvious advantages such as low cost, good reduction and high reactivity [27]. Usually, GO can react with certain molecules of the polymer matrix to form strong interfaces due to the functional groups on the base and edges [28]. To solve the migration problem of small molecules, using GO as modification materials has attracted a lot of attention from researchers in recent years [28,29,30]. There are two methods for employing GO as an anti-migration material. One is to coat the substrate with GO. Su et al. [28] painted a graphene film on the surface of PET to test its resistance to oil and water, and the results showed excellent anti-migration performance. Another way is to add GO to the matrix material to produce composites with anti-migration properties. Khaki et al. [29] introduced GO nanosheets into PLLA materials and they found that graphene promoted the crystallization of PLLA and that the migration resistance of the composites was greatly improved. Damaril et al. [30] prepared graphene/polyurethane composites using modified graphene and obtained a 50% increase in resistance to water vapor migration.

In this study, we combined the excellent barrier effect of GO with the cross-linked structure of HTPB to prepare an anti-migration composites liner. The IGO filled HTPB composites (IGH) with a cross-linked structure were prepared by cross-linking IGO with HTPB chains. The contribution of IGO to the anti-migration and ablation resistance performance of the IGH composite liner were mainly investigated. Moreover, the mechanical properties, crosslink density, and filler dispersion of HTPB was also evaluated.

## 2. Experimental

The section of detailed materials and synthesis of IGO could be observed in the electronic Appendix A.

### 2.1. Preparation of IGO/HTPB Composite Liner

As illustrated in Figure 1, the preparation method of the IGO/HTPB (IGH) composite is as follows: (1) A quantity of IGO (0, 0.3, 0.5, 0.7 wt%) was uniformly dispersed and mixed with HTPB prepolymer by mechanical stirring. (2) The mixture was placed in an oil bath at 80 °C for 3 h to obtain the IGO cross-linked polymer network. (3) IPDI was introduced at a curing factor *R* = 1.3, and then the well-mixed mixture was poured into a 100 mm × 100 mm × 3 mm mold. (4) Finally, the mold was placed in a vacuum oven, cured at 80 °C for 36 h to obtain the IGH composites.

### 2.2. Characterization

Fourier transform infrared spectroscopy (FTIR) (NICOLETIS10, Thermo Fisher Scientific, Waltham, MA, USA) was performed in the wavenumber range of 4000–400 cm^−1^. Laser Raman spectra were measured on a Renishaw in Via spectrometer, with a laser excitation of 532 nm. X-ray diffraction (XRD) patterns were measured using a D8 Advance diffractometer (Bruker-AXS D8 Advance) with Cu Kα radiation (λ = 0.154 nm) at 40 kV and 40 mA. The microstructure was examined by Field Emission Scanning Electron Microscope (FESEM, JSM-7800F PRIME) and transmission electron microscopy (TEM, TECNAI G2 20 LaB6). The mechanical properties were measured using a electronic universal testing machine (SHIMADZU AGS-500N). The crosslink density was measured by soxhlet extraction method. Dynamic mechanical analysis (DMA) measurements were conducted on a VA3000 (01-dB Corporation, France) in the tension mode at a frequency of 10 Hz in the range of −100 °C to 60 °C at a heating rate of 5 °C min^−1^. X-ray photoelectron spectroscopy (XPS) was performed by an X-ray photoelectron spectrometer (PHI QUANTERA II) with an Al Kα X-ray source. The anti-migration performance was measured by the immersion method.

## 3. Results and Discussion

### 3.1. Structural Characterization of IGO

The FTIR spectra of GO and IGO are presented in Figure 2a. The characteristic peaks of GO appeared at 3400, 1620 and 1734 cm^−1^, which correspond to the stretching vibrations of -OH, C=C and C=O, respectively. After being functionalized by IPDI, IGO showed new absorption peaks at 1640 cm^−1^ (C=O stretching vibration), 1391 cm^−1^ (C-N stretching vibration), and 658 cm^−1^ (N-H bending vibrations), confirming the formation of -NH-CO-, which were attributed to reaction between isocyanate groups (-NCO) of IPDI and carboxylic or hydroxyl groups of GO. In addition, there is a characteristic peak at 2256 cm^−1^ (-NCO), indicating the successful introduction of -NCO on GO surface.

The XRD patterns of GO and IPDI-GO are shown in Figure 2b, which shows the structural evolution. There is a sharp and narrow characteristic diffraction peak of GO (110) at 2θ = 10.58° and the calculated interplanar spacing is 0.834 nm. After modification, IGO presents a diffraction peak at 9.14° and the peak is weaker. There is a sharp and narrow characteristic diffraction peak of GO (110) at 2θ = 10.58° and the calculated interplanar spacing is 0.834 nm. After modification, IGO presents a diffraction peak at 9.14° and the peak is weaker. According to 2*d*sinθ = *n*λ, the interlayer spacing of IPDI-GO increased to 0.966 nm, which is due to the intercalation modification of IPDI further expanding the interplanar spacing of GO. In summary, analysis of the IPDI-GO by XRD revealed IPDI is grafted onto GO.

As can be seen in Figure 2c, both GO and IGO show absorption peaks near 1348 cm^−1^ and 1590 cm^−1^, which correspond to the absorption peaks of sp^3^ hybridized or disordered carbon atoms representing defects in the graphene lamellae (D peaks) and the stretching vibration peaks of sp^2^ hybridized carbon atoms representing order (G peaks), respectively. The ratio of peak intensities (I_D_/I_G_) measures the degree of crystallization of graphene [25], and a larger I_D_/I_G_ ratio indicates more defects in the graphene lamellar structure. The I_D_/I_G_ ratio of graphene oxide was 0.89, and after IPDI modification, the I_D_/I_G_ ratio increased to 1.00 with more defects, indicating that the IPDI modification process destroys the original graphene lamellar structure, which is beneficial to improve its compatibility with the HTPB substrate.

Figure 3a–d shows the microscopic appearance of GO and IGO nanosheets, respectively. As shown in Figure 3b, the prepared GO exhibited stacked layers and irregular sheets. After intercalation modification of IGO with IDPI, the lamellae are larger and stacked together, which can more effectively prevent the passage of small molecules, thereby improving the migration resistance of IGH liner. TEM images of GO and IGO are shown in Figure 3e,f; the GO sheet has a smooth surface and visible wrinkles, similar to tulle. Compared with GO, IGO has less transparency and a more three-dimensional structure. The dispersibility is also improved due to the IDPI modification, which hindered π-π stacking between GO flakes.

The binding energy of GO and IGO are shown in Figure 4a and the element contents are listed in Table 1. There is an N 1s peak at 400.3 eV of IGO spectrum; the data in Table 1 show that the N element content reached 10.5%, which confirms the reaction between -NCO of IPDI and carboxylic or hydroxyl groups on the GO surface. In addition, IGO exhibits relatively higher intensity ratio of C1s and O1s peaks due to the higher C/O ratio of IPDI and the O content decreased from 34.6% to 14.1%, indicating the existence of IPDI molecules grafted onto the GO surface. The C1s spectra of GO and IGO are shown in Figure 4b,c, respectively. We can see that GO exhibits six different binding states: C=C (283.9 eV), CO- H (284.7 eV), C=O (286.6 eV), COC (288.1 eV), and C(=O)-O (291.1 eV) [31]. There is a new C-N bond peak at 286.1 eV in the IGO spectrum. The C-O and C-O-C peaks were weakened, and the C-O-H peak disappeared, indicating that IPDI fully reacted with carboxylic or hydroxyl groups of GO. Figure 4d shows the N 1s spectrum of IGO. During grafting, the -NCO group at one end of the IPDI benzene ring reacted with oxygen-containing functional groups, and the -NCO group at the other end remained on the GO surface. There is a new peak at 398.9 eV (-NHCOO) in the spectrum, indicating that IPDI reacted with GO to form a covalent bond. The binding peak of -NCO at 400.7 eV indicates that the GO surface was successfully grafted with isocyanate groups.

### 3.2. Microscopic Morphology of Composites Liner

The dispersion of the filler in the matrix greatly affected the performance of the composites. We use SEM to further investigate the dispersion of the IPDI in the composites. Figure 4a–c shows the SEM images of IGH composites with different IGO content. Some layered structures can be seen in the fracture surface with the increasing IGO content; the appearance of cracks on the fracture surface means the enhancement of interfacial interaction, indicating an enhance in the hardness and rigidity of the composites. As shown in Figure 5, IGO particles uniformly dispersed in the matrix. Consequently, the homogenously dispersed IGO could create a positive effect, restricting the pathway of plasticizer diffusion.

### 3.3. Crosslink Density Analysis

Crosslink density is an essential characteristic of crosslinked polymers, and plays an important role in plasticizer migration. Usually, polymers with high cross-linkage are more tightly polymerized, firm and durable, with higher density and fewer internal voids, and thus are more resistant to plasticizer migration. In this part, we use soxhlet extraction to test the crosslink density of liners. The crosslink densities of IGH composites are shown in Figure 6.

Among the IGH composites liners, the crosslink density of pure HTPB is the lowest, and the crosslink density of the IGH composite liner was increased with the IGO. IGH-0.5 has the highest crosslink density of 96.43%, increased by 5.71%. As shown in Figure 7, the -OH group of HTPB chain and the -NCO of IPDI form a stable cross-linked structure during the curing and molding process. There are many oxygen-containing groups on the surface of GO, and some isocyanate groups on the curing agent react with these hydroxyl groups. For IGH liner composite, IGO is modified by IPDI, so that there is a large number of -NCO groups that can react with the hydroxyl groups of the HTPB chains to form a multi-crosslinked structure. Therefore, IGO is no longer simply dispersed in the matrix, but acts as a cross-linking agent to form a cross-linked structure with HTPB, so that the crosslink density is increased.

### 3.4. Ablation Resistance

The composite liner also needs to have good ablation resistance. Oxy-acetylene flame flow was used as the heat source to test the ablation performance of the composite cladding specimens, and the results are shown in Table 2. R_L_ is the line ablation rate, R_L_ is the average line ablation rate. As shown in Table 2, with the increase of IGO content, R_L_ of different samples decreases gradually. HTPB has the worst ablation resistance, with a R_L_ of 0.652 mm/s; IGH-0.7 has the best ablation resistance, with a R_L_ of 0.431 mm/s, a decrease of 34.10% compared with the former. The improved ablation resistance was due to the high thermal stability of IGO, which can improve the thermal stability of the polymer matrix. In addition, it is dispersed in the matrix in a lamellar structure, forming curved channels, which slows down the infiltration of oxygen from the surface to the interior and the escape of gases generated during combustion, as shown in Figure 8, thus reducing the burning rate and improving the ablation resistance. The crosslink also contributed to ablation resistance because of the composites become more dense, blocking the escape of gases.

### 3.5. Mechanical Properties

DMA analysis was used to study the enhancement of IGO on HTPB and the interfacial interaction between IGO and HTPB chains. Figure 9 showed the dynamic mechanical performance of IGH composites. The G′ of IGH composites increased with the increase of IGO content. The result showed a significant decrease in mobility of the HTPB chain around IGO nanosheets owing to the crosslinking as well as an efficient load transfer from the matrix chain to GO nanosheets [26]. Figure 9b shows the loss factor (tan δ) vs. temperature for all IGH composites. The peak tan δ of IGH composites is lower than that of pure HTPB as the mobility of HTPB molecules is limited by the filler network of IGO. Therefore, the interfacial interactions between IGO and HTPB molecules were further enhanced, which is beneficial to the improvement of anti-migration properties.

The tensile properties of IGH composites are shown in Figure 10c. The tensile strength and elongation at break of pure HTPB are 1.11 MPa and 210.40%, respectively. IGH-0.5 shows the best performance in tensile strength and elongation at break, exhibiting an increment up to 24% and 54% compared with that of pure HTPB, respectively. It is important to note that both tensile strengths and elongation at break are enhanced with the introduction of IGO. The tensile strength and break elongation of IGH-0.7 are shown to be decreased probably due to the aggregation [27], but still higher than the pure HTPB. The increase in mechanical properties not only depends on the type of filler, but is also affected by surface modification, filler dispersion, and filler/matrix interface interaction. IGO is uniformly dispersed in the matrix, greatly improving the interfacial bonding. In addition, the cross-link between IGO and HTPB chains are also responsible for the reinforcement mechanical properties.

### 3.6. Anti-Migration Performance

Plasticizers increase the flexibility of inhibitor and improve the mechanical properties of propellants. However, during the storage of a solid rocket motor, small molecules such as NG will migrate to the inhibitor, which may damage the solid engine and cause major accidents. However, due to the limitation of experimental conditions, DOS was used for solution immersion experiments to indirectly compare the anti-migration performance of the composites. In this paper, the anti-migration performance is measured by the immersion method. Figure 10 shows the concentration of DOS that migrated to the composites at different temperatures, which represent the anti-migration performances of composites. As show in Figure 10, DOS will dynamically migrate driven by the concentration gradient. In the early stage of impregnation, the migration of DOS increased rapidly and then slowed down over time because the migration reached equilibrium.

Figure 10a shows the migration curves of DOS in IGH composites liner at 25 °C. The addition of IGO significantly improved the anti-migration performance. With the increase of IGO content, the migration equilibrium concentration of DOS decreases, and the anti-migration properties of the IGH composites liner on DOS are gradually increased. When 0.5 wt% IGO is introduced, IGH-0.5 achieved the lowest migration equilibrium concentration. The equilibrium concentration of DOS migration is 53.91%, which was 12.60% lower compared to the pure HTPB. The improved anti-migration performance is attributed to the excellent resistance to small molecule penetration of graphene. IGO is uniformly dispersed in the matrix and acts as a barrier to the migration of DOS. The lamellar structure blocks the migration path of the DOS, thus increasing the migration resistance of the liner. In addition, the cross-link density is increased with the introduction of IGO. IGO is no longer simply dispersed in the matrix, but acts as a cross-linking agent to form a cross-linked structure with HTPB. The cross-linked structure thus becomes more compact and more effective in blocking the structural gaps through which DOS passes, ultimately achieving a reduction in migration. We can find that the migration resistance of IGH-0.7 is slightly decreased because of the uneven dispersion, which is due to the agglomeration of IGO in the matrix. However, the migration resistance of IGH-0.7 is better than that of HTPB, indicating IGO has significant effect on migration resistance of the composite liner.

We also studied the anti-migration properties of samples at different temperatures. It can be seen from the figure that for the same liner sample, the migration equilibrium concentration of DOS increased with the temperature, and equilibrium can be achieved in a short time. This is because the increased molecular motion at high temperatures increases the intermolecular distance and the free volume of the composites, thereby expanding the migration channel of DOS. Thus, the anti-migration performance is decreased. The migration process is accelerated by the increased molecular motion of DOS at high temperature, which also affects the migration resistance.

### 3.7. Migration Kinetics and Thermodynamics

The diffusion coefficient (DC) is an essential kinetic parameter used to describe the speed of the migration process [31]. We processed the data obtained by the immersion method according to Fick’s second law (1) and calculated the diffusion coefficient through linear fitting. Linear fitting of the DOS migration coefficient in the IGH composite at different temperatures is shown in Figure 11, and the diffusion coefficients are given in Table 3.
(1)$$D=\pi {\left(\frac{m_{t}}{m_{0}}-1\right)}^{2}{\left(\frac{\rho_{i}d_{i}}{2\rho }\right)}^{2}\frac{1}{t}$$
where *m_t_* represents the mass of the diffused substance sheet including the absorbed substance, *m*_0_ is the primal mass of the sheet, *ρ_i_* indicates the density of the sheet, *ρ* is the density of the immersion liquid; *d_i_* represents the thickness of the sheet.

The higher the diffusion coefficient, the faster DOS diffuses, resulting in an increase in migration. It is worth noting that the order of magnitude of the migration diffusion coefficients decreases from 1 × 10^−3^ to 1 × 10^−4^ after the introduction of IGO, indicating a significant decrease in the migration rate of DOS in the IGH liner. It can be seen from Figure 10 that as temperature increases, the diffusion reaches equilibrium within a shorter time and at a higher migration concentration. At the same time, the migration diffusion coefficients increase with increasing temperature because plasticizer molecules move violently at high temperatures, resulting in an increase in the diffusion rate. In addition, prolonged high temperature also leads to the aging of HTPB, and the disruption of the cross-linked structure also reduces the migration resistance. The change of migration diffusion coefficients is opposite to the trend of crosslink density change, implying that crosslink density has a significant impact on plasticizer migration.

The migration activation energy (*E_a_*) of DOS can be calculated using the Arrhenius equation [32] and the results of a linear fit combining the migration diffusion coefficient and temperature are shown in Figure 12. The calculated values of *E_a_* are listed in Table 4. The *E_a_* of IGH-0.5 is 19.78 kJ/mol, which is higher than that of pure HTPB, confirming that the introduction of IGH enhances the anti-migration properties. At the same time, the crosslink density of the composite increased after the IGH introduction. Combined with the above data, this can indicate that IGO crosslinked with the HTPB matrix formed a tighter structure making it difficult for the plasticizer molecules to migrate.
(2)$$\mathrm{lnD}=\mathrm{lnA}-\frac{E_{a}}{RT}$$
in which D is the migration coefficient, A represents the empirically derived constant, *E_a_* is the activation energy of migration, *R* is the universal molar constant, *T* represents the temperature.

## 4. Conclusions

In this work, we obtained IPDI-GO by modifying graphene oxide using IPDI and subsequently cross-linked it with HTPB to prepare composite liner. The composite liner has a cross-linked structure due to the reaction of -NCO and -OH group. The investigation of plasticizer migration at different temperatures showed that the introduction of IGO prevented plasticizer migration into HTPB composites. IGH-0.5 showed the best anti-migration effect. Compared to pure HTPB, the migration resistance increased by 18.94%, 16.33% and 15.34% at 25 °C, 60 °C and 90 °C, respectively, and the migration coefficient was reduced by an order of magnitude. IGO was introduced as a reinforcing and cross-linking agent, providing good migration resistance and ablation resistance to the composite liner. In conclusion, this study indicates that the cross-linked structure is beneficial to improve the comprehensive performance of the composites.

## Figures and Tables

**Figure 1 polymers-14-03315-f001:**
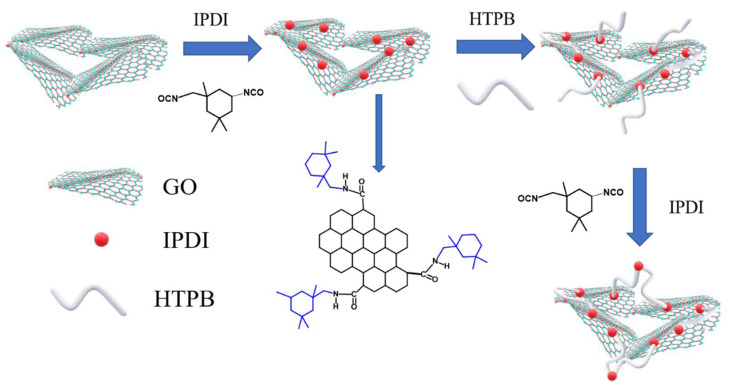
The preparation of IGH composites.

**Figure 2 polymers-14-03315-f002:**
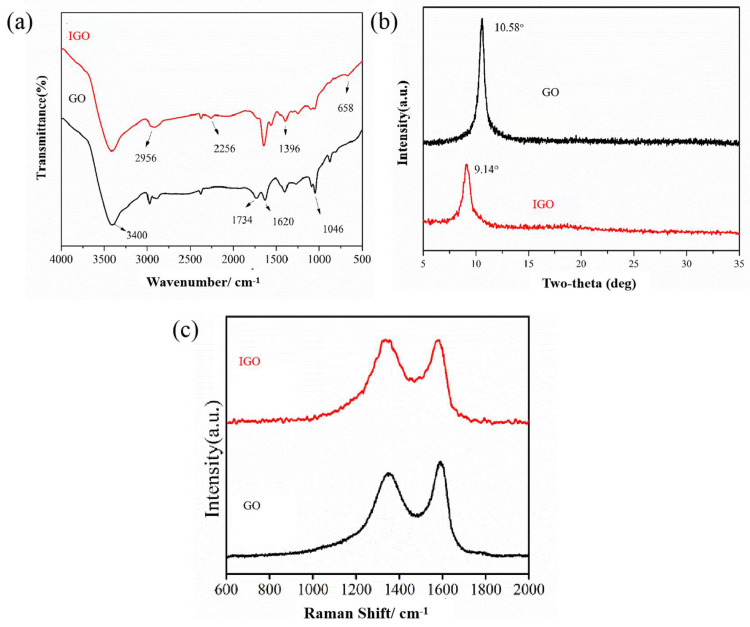
FTIR spectra (**a**), XRD spectra (**b**) and Raman spectra (**c**) of GO and IGO.

**Figure 3 polymers-14-03315-f003:**
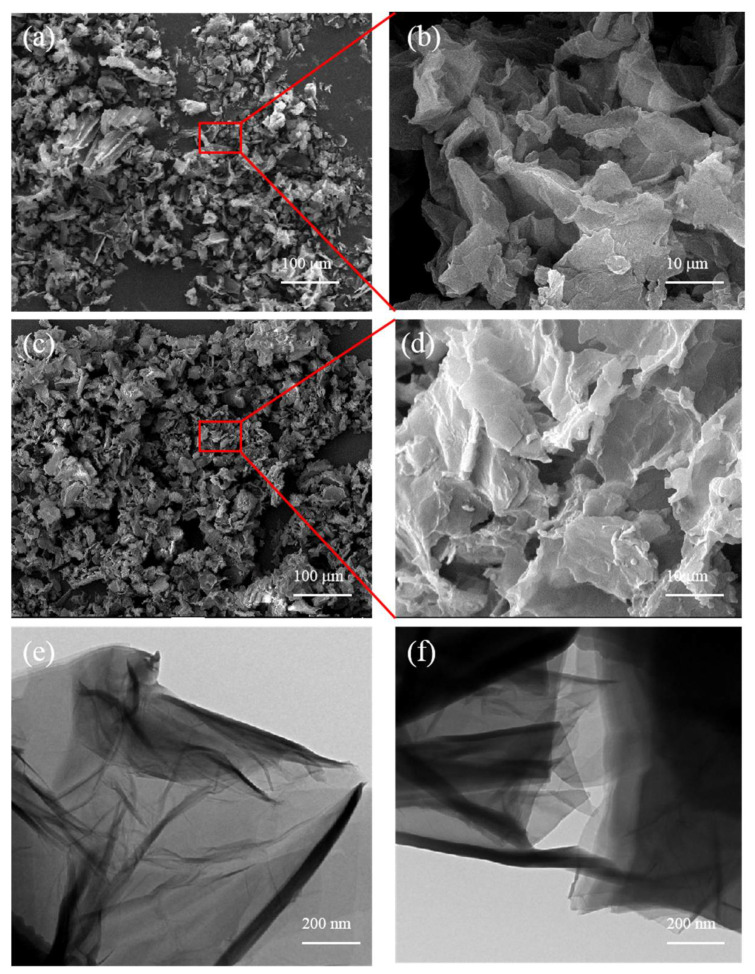
SEM images of (**a**,**b**) GO, (**c**,**d**) IGO; TEM images of (**e**) GO and (**f**) IGO.

**Figure 4 polymers-14-03315-f004:**
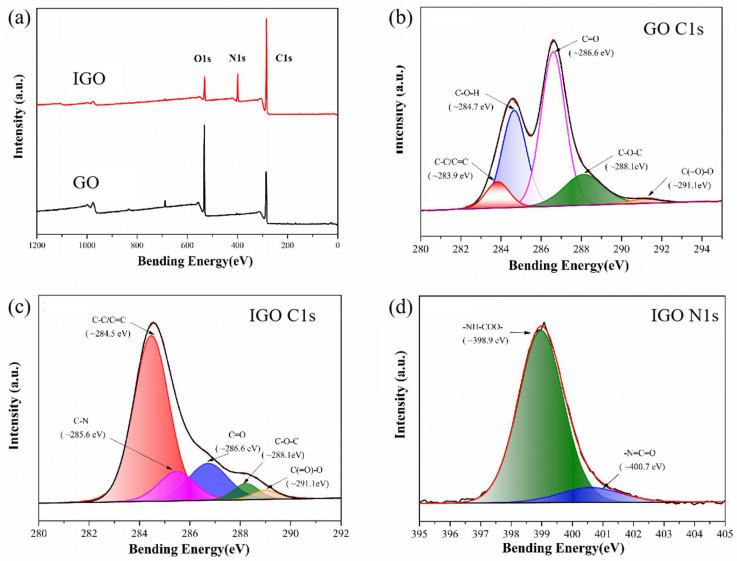
(**a**) XPS survey spectra of GO and IGO. (**b**,**c**) High-resolution C1s XPS spectra of GO and IGO, respectively. (**d**) High-resolution N1s XPS spectra of IGO.

**Figure 5 polymers-14-03315-f005:**
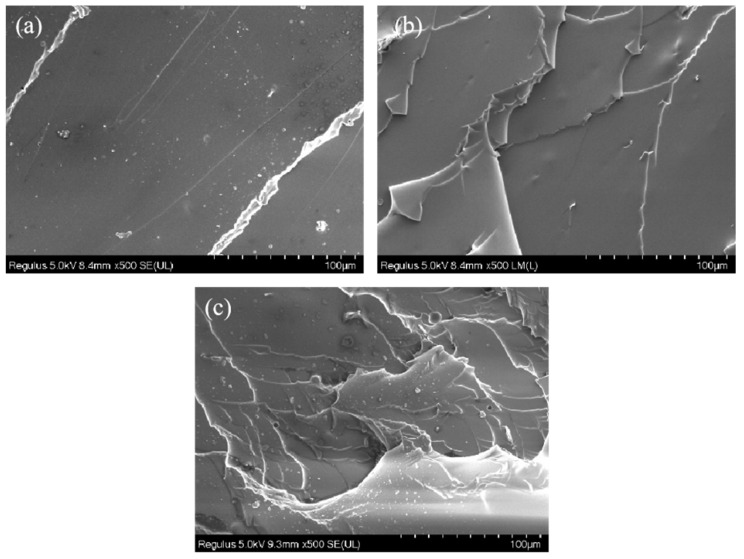
(**a**–**c**) SEM images of the fracture surface of IGH composite liners with 0.3, 0.5 and 0.7 wt% IGO, respectively.

**Figure 6 polymers-14-03315-f006:**
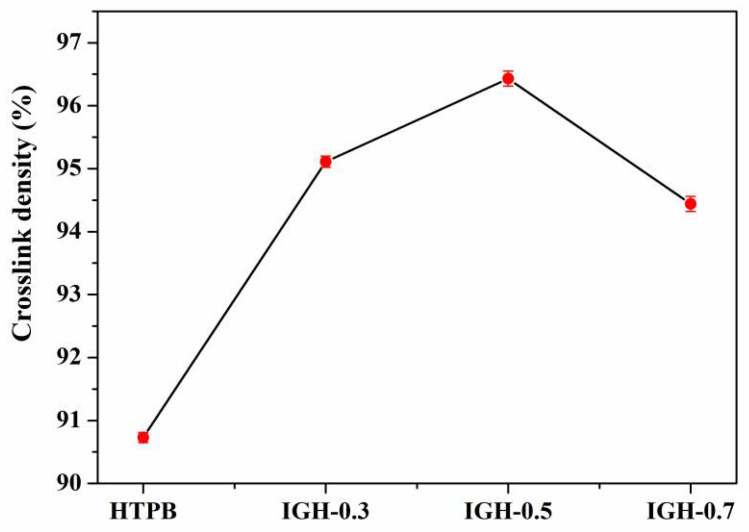
Crosslink density of samples with different IGO contents.

**Figure 7 polymers-14-03315-f007:**
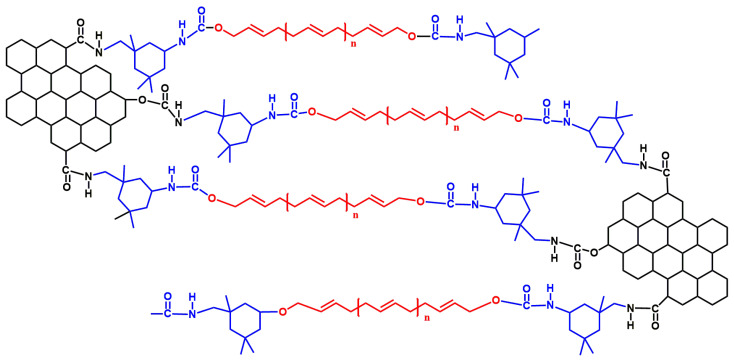
Cross-link reaction between IPDI, HTPB and IGO.

**Figure 8 polymers-14-03315-f008:**
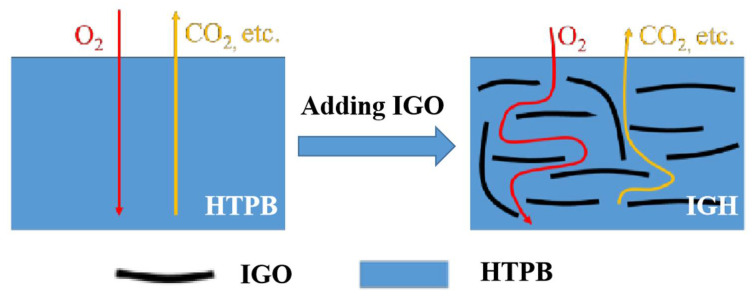
Diagram of the barrier mechanism.

**Figure 9 polymers-14-03315-f009:**
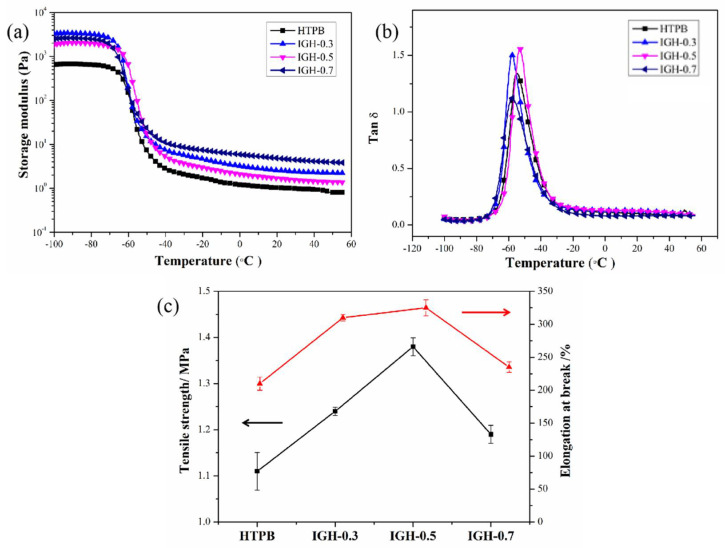
Storage modulus (**a**) and Tan (δ) (**b**) vs. temperature of IGH composites, (**c**) Mechanical properties of IGH composites with different IGO contents.

**Figure 10 polymers-14-03315-f010:**
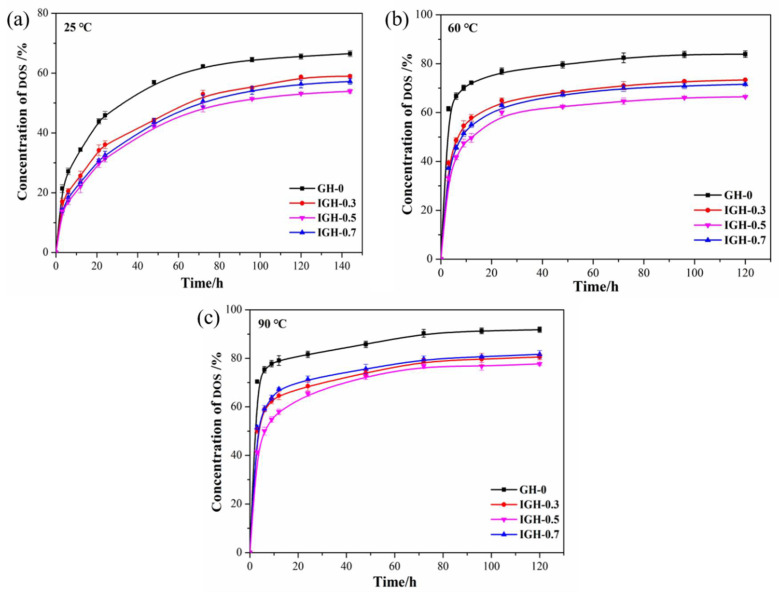
The migration curves of DOS in the IGH composites at different temperatures: (**a**) 30 °C, (**b**) 60 °C, and (**c**) 90 °C.

**Figure 11 polymers-14-03315-f011:**
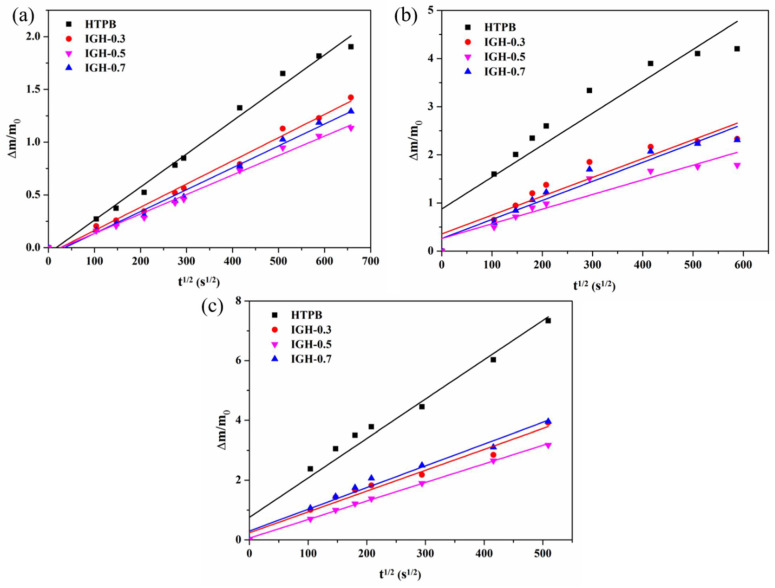
Linear fitting of the DOS migration coefficient in the IGH composite at different temperatures: (**a**) 25 °C, (**b**) 60 °C, and (**c**) 90 °C.

**Figure 12 polymers-14-03315-f012:**
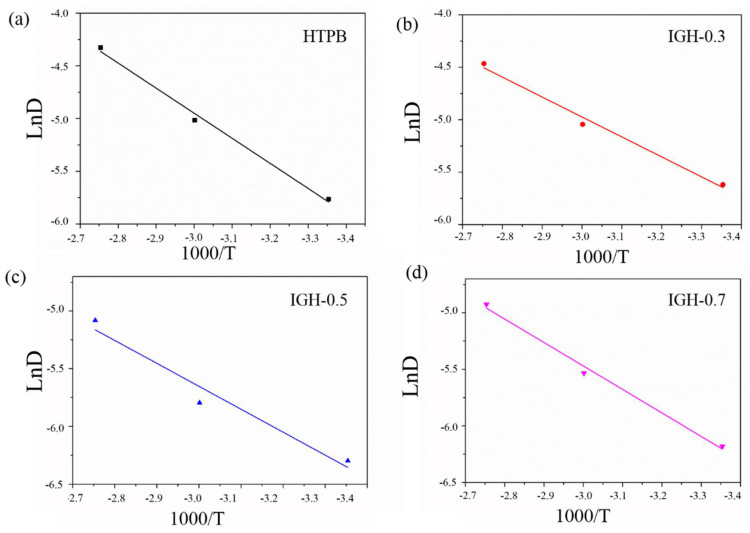
The plots of lnD against 1000/T at different temperatures for different samples: (**a**) HTPB, (**b**) IGH-0.3, (**c**) IGH-0.5 and (**d**) IGH-0.7.

**Table 1 polymers-14-03315-t001:** Element contents of GO and IGO.

Sample	C/at%	O/at%	N/at%	C/O Ratio
GO	65.4	34.6	-	1.89
IGO	75.4	14.1	10.5	5.35

**Table 2 polymers-14-03315-t002:** Linear ablation rate curve of coating layer with different IGO contents.

Samples	R_L_/(mm/s)			R_L_/(mm/s)
HTPB	0.651	0.682	0.622	0.652
IGH-0.3	0.496	0.470	0.454	0.473
IGH-0.5	0.464	0.439	0.460	0.454
IGH-0.7	0.429	0.441	0.422	0.431

**Table 3 polymers-14-03315-t003:** The diffusion coefficients of DOS migrating into the IGH composite.

Samples	25 °C		60 °C		90 °C	
	*D* (m^2^ s^−1^)	*R* ^2^	*D* (m^2^ s^−1^)	*R* ^2^	*D* (m^2^ s^−1^)	*R* ^2^
HTPB	3.88 × 10^−3^	0.981	4.66 × 10^−3^	0.968	5.36 × 10^−3^	0.964
IGH-0.3	3.36 × 10^−4^	0.984	3.94 × 10^−4^	0.985	5.13 × 10^−4^	0.973
IGH-0.5	3.12 × 10^−4^	0.987	3.88 × 10^−4^	0.992	4.28 × 10^−4^	0.961
IGH-0.7	3.43 × 10^−4^	0.999	4.67 × 10^−4^	0.977	5.68 × 10^−4^	0.966

**Table 4 polymers-14-03315-t004:** Summary of activation energies of DOS migrate into HTPB composites.

Samples	*E_a_* (kJ/mol)	*R* ^2^
HTPB	15.81	0.993
IGH-0.3	17.25	0.904
IGH-0.5	19.78	0.912
IGH-0.7	16.51	0.971

## Data Availability

Data will be made available on reasonable request.

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
