# Peer review of "Functionalized GO/Hydroxy-Terminated Polybutadiene Composites with High Anti-Migration and Ablation Resistance Performance"

_polymers, 2022, doi:10.3390/polym14163315_

Round 1

Reviewer 1 Report

The manuscript deals with the synthesis of a hydroxy-terminated polybutadiene liner composite material adapted with isocyanate functionalized graphene oxide to inhibit migration of plasticizers.

I have the following main suggestions and comments for this paper:

1) The work should start with an introduction sentence, and context setting. The first sentence of the introduction does not clearly explain the relevance and context of this work. The context should also be presented in a sentence at the start of the abstract. The introduction should also explain why the research is important, which problem one is trying to solve and why it is relevant and useful. A clear overview on the current state of the art should be presented. What have others done so far to solve the problem?

2) The results do not convincingly present that improved lower migration of plasticizer is due to the layered structure of isocyanate functionalized graphene oxide and the dense molecular chain network from the crosslinking. For example, Figure 3d does not show a clear layered structure. More convincing evidence should be provided.

3) The lay-out and presentation is poor. The Tables are too large. Figures captions seem to start with a dot and empty line. This dot and empty line should be removed. Reference to Tables are given as Table. 2 for example, the dot should be removed. Figure 5 does not fit within the margins of the paper.

4) The English language needs to be improved, and type errors should be avoided. There are many in the text, for example, it should be data instead of datas (page 6), needs instead of need (page 8), etc. In the conclusion, the reduction percentages at the different temperatures is missing, etc

5) Reproducibilty. It is not clear from the paper how many repeat experiments and measurements were conducted to ensure reproducibility and reliability of the results.

Author Response

List of Responses:

Dear Editors and Reviewers:

Thank you for your letter and for the reviewer's comments concerning our manuscript entitled  " Functionalized GO / Hydroxy-terminated polybutadiene composites with high anti-migration and ablation resistance performance"(ID:polymers-1870398).Those comments are valuable and helpful for us to improve the quality of our manuscript. After considering the comments carefully, we have revised our manuscript carefully. We hope that the revised version can meet the requirement of the journal. Revised portion are clearly marked by red in the revised manuscript. The main corrections in the paper and the responds to the reviewer's comments are as flowing

Responds to the reviewer's comments:

Reviewer #1: Comments

The manuscript deals with the synthesis of a hydroxy-terminated polybutadiene liner composite material adapted with isocyanate functionalized graphene oxide to inhibit migration of plasticizers.

I have the following main suggestions and comments for this paper:

  • The work should start with an introduction sentence, and context setting. The first sentence of the introduction does not clearly explain the relevance and context of this work. The context should also be presented in a sentence at the start of the abstract. The introduction should also explain why the research is important, which problem one is trying to solve and why it is relevant and useful. A clear overview on the current state of the art should be presented. What have others done so far to solve the problem?

Response: Thank you for your advices. We have revised the background part based on your suggestion, all changes are highlighted in red. For example: The sentence at the start of the abstract:The migration of plasticizers such as nitroglycerin seriously affects the storage and working safety of rocket systems. We have revised the Introduction to enrich the relevance and context of our work.The migration of large amounts of NG may damage the solid engine and cause major accidents. how to propose effective methods to prevent small molecule migration in propellant in terms of material design and mechanism is a key scientific problem.Many works have attempted to prevent the migration of NG, the most convenient method is using a polymer with strong anti-migration ability as the inhibitor [2,14]. EPDM rubber has strong resistance to small molecule and Lu ZH et al. used it to resist dioctyl sebacate (DOS) migration. The addition of inorganic filler is also beneficial to the an-ti-migration performance, inorganic filler can increase the density of the coating layer to play the role of anti-migration [15]. In addition, setting a barrier layer between the propel-lant and the inhibitor can significantly reduce the migration of NG[16].

  • The results do not convincingly present that improved lower migration of plasticizer is due to the layered structure of isocyanate functionalized graphene oxide and the dense molecular chain network from the crosslinking. For example, Figure 3d does not show a clear layered structure. More convincing evidence should be provided.

Response:Thanks for your suggestion. Figure 3d shows the microscopic appearance of IGO nanosheets, the lamellae are larger and stacked together. The improved anti-migration performance is attributed to the excellent resistance to small molecule penetration of graphene, IGO is uniformly dispersed in the matrix and acts as a barrier to the migration of DOS. The layered structure we refer to is the state in which IGO is dispersed in the HTPB matrix, as shown in the Figure 8.The lamellar structure blocks the migration path of the DOS thus increasing the migration resistance of the liner. In addition, with the introduced of IGO, the cross-link density is increased. The XPS data of IGO are shown in Figure 5, according to the XPS data, the -NCO group at one end of the IPDI benzene ring reacted with oxygen-containing functional groups, and the -NCO group at the other end remained on the GO surface. The newly appeared bound state -NHCOO (398.9 eV) in the spectrum indicated that IPDI reacted with GO to form a covalent bond. The binding peak of -NCO at 400.7 eV indicates that the GO surface was successfully grafted with isocyanate groups, revealing the  reactive ability with HTPB chains. IGO is no longer simply physically mixed as a filler, but is embedded in the matrix through group bonding. The cross-linked structure thus becomes more compact and more effective in blocking the network gaps through which plasticizer molecules pass, ultimately achieving a reduction in migration.

  • The lay-out and presentation is poor. The Tables are too large. Figures captions seem to start with a dot and empty line. This dot and empty line should be removed. Reference to Tables are given as Table. 2 for example, the dot should be removed. Figure 5 does not fit within the margins of the paper.

Response: Thank you for your advice. We are sorry for our negligence, we have made correction according to your suggestion and marked them in red.

  • The English language needs to be improved, and type errors should be avoided. There are many in the text, for example, it should be data instead of datas (page 6), needs instead of need (page 8), etc. In the conclusion, the reduction percentages at the different temperatures is missing, etc

Response: We thank the reviewer for pointing this out. We have carefully checked the manuscript again, and some grammatical errors have be corrected and highlighted in red in the revised-manuscript.

  • It is not clear from the paper how many repeat experiments and measurements were conducted to ensure reproducibility and reliability of the results.

Response: Thanks for pointing this out. The experiments and tests mentioned in the text were performed in multiple repetitions, eg: linear ablation rate and crosslink density are the average of three measurements. Tensile properties are the average of five measurements with added error. The immersion tests used to measure the anti- migration performance were repeated three times for each group of samples.

Once again, thanks very much for your comments and suggestions.

Reviewer 2 Report

This paper presented aspects regarding the development of hydroxy-terminated polybutadiene (HTPB) liner composites with the cross-linked structure were prepared by cross-linking isocyanate functionalized graphene oxide (IGO) with HTPB to prevent the migration of high energy plasticizers in the propellant. The objectives of the study were to evaluate the effect of functionalized GO on the crosslink density, and filler dispersion on the HTPB mechanical properties. Also, the anti-migration and adhesion properties of the composite liner were studied.

The paper is original and relevant in the field. The paper is clear presented and well structured.

The conclusions are in agreement with the evidence and arguments presented.

The references are up to date and in accordance with the subject approached. The following reference could be included: Hao Li, Wenjia Jiang, Yanan Zhang, Zhehong Lu, Yubing Hu, Chuan Xiao, Tengyue Zhang, Fengya She, Solid propellant liner with high anti-migration and strong adhesion based on isocyanate functionalized graphene oxide and hydroxy-terminated polybutadiene, J Mater Sci (2022) 57:14413–14429.

Please make the following corrections:

Page 6-7, Table 1 – “IGO” instead of “TGO”

Page 10, line 11 – “As show in Fig.8,…” or is Fig.10?

Page 11, Figure 9 - “IGO contents” instead of “TGO contents”

Page 12, Figure 10 - “HTPB inhibitor coating” instead of “EPDM inhibitor coating”

Page 14, Table 4 - “HTPB composites” instead of “EPDM composites”

Page 14, Figure 12 – define a, b, c, d

Page 14 – verify the sentence: In which is the migration coefficient, A represents the empirically derived constant, Ea is the activation energy of migration, R is the universal molar constant, T represents the temperature.”, maybe “Where D is the migration coefficient, ….”

I consider that the paper is of interest and it can be considered for publication.

Author Response

List of Responses:

Dear Editors and Reviewers:

Thank you for your letter and for the reviewer's comments concerning our manuscript entitled " Functionalized GO / Hydroxy-terminated polybutadiene composites with high anti-migration and ablation resistance performance"(ID:polymers-1870398).Those comments are valuable and helpful for us to improve the quality of our manuscript. After considering the comments carefully, we have revised our manuscript carefully. We hope that the revised version can meet the requirement of the journal. Revised portion are clearly marked by red in the revised manuscript. The main corrections in the paper and the responds to the reviewer's comments are as flowing

Responds to the reviewer's comments:

This paper presented aspects regarding the development of hydroxy-terminated polybutadiene (HTPB) liner composites with the cross-linked structure were prepared by cross-linking isocyanate functionalized graphene oxide (IGO) with HTPB to prevent the migration of high energy plasticizers in the propellant. The objectives of the study were to evaluate the effect of functionalized GO on the crosslink density, and filler dispersion on the HTPB mechanical properties. Also, the anti-migration and adhesion properties of the composite liner were studied.

The paper is original and relevant in the field. The paper is clear presented and well structured.

The conclusions are in agreement with the evidence and arguments presented.

The references are up to date and in accordance with the subject approached. The following reference could be included: Hao Li, Wenjia Jiang, Yanan Zhang, Zhehong Lu, Yubing Hu, Chuan Xiao, Tengyue Zhang, Fengya She, Solid propellant liner with high anti-migration and strong adhesion based on isocyanate functionalized graphene oxide and hydroxy-terminated polybutadiene, J Mater Sci (2022) 57:14413–14429.

Please make the following corrections:

Page 6-7, Table 1 – “IGO” instead of “TGO”

Page 10, line 11 – “As show in Fig.8,…” or is Fig.10?

Page 11, Figure 9 - “IGO contents” instead of “TGO contents”

Page 12, Figure 10 - “HTPB inhibitor coating” instead of “EPDM inhibitor coating”

Page 14, Table 4 - “HTPB composites” instead of “EPDM composites”

Page 14, Figure 12 – define a, b, c, d

Page 14 – verify the sentence: “In which is the migration coefficient, A represents the empirically derived constant, Ea is the activation energy of migration, R is the universal molar constant, T represents the temperature.”, maybe “Where D is the migration coefficient, ….”

I consider that the paper is of interest and it can be considered for publication.

Response: Thank you very much for your constructive comments and suggestions on our manuscript.We are sorry for our negligence, we have made correction according to your suggestion and marked them in red. Your suggested reference citation is as follows:

[22] Li H, Jiang W, Zhang Y, et al. Solid propellant liner with high anti-migration and strong adhesion based on isocyanate-functionalized graphene oxide and hydroxy-terminated polybutadiene. J. Mater. Sci. 2022, 57, 14413.